# Reinterpretation of the results of randomized clinical trials

**Farrokh Habibzadeh** [ORCID] *

Global Virus Network, Middle East Region, Shiraz, Iran

* Farrokh.Habibzadeh@gmail.com

## Abstract

### Background

Randomized clinical trials (RCTs) shape our clinical practice. Several studies report a mediocre replicability rate of the studied RCTs. Many researchers believe that the relatively low replication rate of RCTs is attributed to the high p value significance threshold. To solve this problem, some researchers proposed using a lower threshold, which is inevitably associated with a decrease in the study power.

### Methods

The results of 22 500 RCTs retrieved from the Cochrane Database of Systematic Reviews (CDSR) were reinterpreted using 2 fixed p significance threshold (0.05 and 0.005), and a recently proposed flexible threshold that minimizes the weighted sum of errors in statistical inference.

### Results

With $p < 0.05$ criterion, 28.5% of RCTs were significant; $p < 0.005$, 14.2%; and $p <$ flexible threshold, 9.9% (2/3 of significant RCTs based on $p < 0.05$ criterion, were found not significant). Lowering the p cut-off, although decreases the false-positive rate, is not generally associated with a lower weighted sum of errors; the false-negative rate increases (the study power decreases); important treatments may be left undiscovered. Accurate calculation of the optimal p value thresholds needs knowledge of the variance in each study arm, *a posteriori*.

### Conclusions

Lowering the p value threshold, as it is proposed by some researchers, is not reasonable as it might be associated with an increase in false-negative rate. Using a flexible p significance threshold approach, although results in a minimum error in statistical inference, might not be good enough too because only a rough estimation may be calculated *a priori*; the data necessary for the precise computation of the most appropriate p significance threshold are only available *a posteriori*. Frequentist statistical framework has an inherent conflict. Alternative

**Data Availability Statement:** The original data set is publicly available from OSF (https://doi.org/10.17605/OSF.IO/XJV9G).

**Funding:** The author(s) received no specific funding for this work.

**Competing interests:** The author has declared that no competing interests exist.

methods, say Bayesian methods, although not perfect, would be more appropriate for the data analysis of RCTs.

## Introduction

Heavily relying on systematic reviews of randomized clinical trials (RCTs), evidence-based medicine is currently considered the most appropriate way to practice medicine. A recent article that examined more than 20 000 RCTs retrieved from the Cochrane Database of Systematic Reviews (CDSR), arguably the most comprehensive database of evidence on medical interventions, reports a mediocre replicability rate of the studied RCTs; the probability that a replication of the studied RCTs with p values ranging from 0.001 to 0.05, also ends with a significant p value (conventionally, a $p < 0.05$) with an observed effect in the same direction is just a little bit more than 40% [1]. Many researchers believe that the relatively low replication rate of RCTs is attributed to the high p value significance threshold (PST), arbitrarily set to 0.05, and that the PST should be chosen reasonably, not arbitrarily.

The idea of using the p value is often credited to Karl Pearson who outlined its basic framework in 1900. However, it seems that it was John Arbuthnot who first employed the idea to test a hypothesis in 1710 [2]. Arbuthnot reviewed the number of male and female neonates born in London between 1629 and 1710 and found that the number of males was consistently higher than that of females over 82 studied years. He calculated the probability of observing such a consistent male excess by chance alone and found that it would be an incredibly small value ($0.5^{82} \approx 2.07 \times 10^{-25}$), if the birth rates of males and females were really equal. He then concluded that the observed difference was highly unlikely to occur at random under the hypothesis that the birth rates are equal, and that the male birth rate was truly higher than the female birth rate [2, 3]. In 1925, when Ronald A. Fisher formalized Pearson's idea of the p value, he arbitrarily proposed setting the PST at 0.05 (*i.e.*, a 1 in 20 chance) [4].

Over the past years, several simulation studies have shown that a significant p value (conventionally, a $p < 0.05$) can easily be attained merely by chance, which implies that many of the "significant results" obtained in RCTs could really be false-positive [5–7]. To address this problem, some investigators have proposed setting the PST at a lower value [8–11]. For instance, Ioannidis has proposed to lower the PST from the conventional value of 0.05 to 0.005 [11], a proposal that has also been supported by Benjamin, *et al* [10]. McCloskey and Michaillat have also proposed to decrease the PST from 0.05 to 0.01 [12]. All these proposals for a smaller but fixed PST, nonetheless, suffer from the very same problem that the conventional PST of 0.05 has faced with—there is no logical reason behind choosing a constant PST. And, that is probably why none of the proposals has so far gained universal acceptance.

There is a trade-off between the false-positive and false-negative rates—any attempt to decrease the false-positive rate (*e.g.*, by decreasing the PST) is associated with an increase in the false-negative rate, which is tantamount to a decrease in the study power [13], the probability that a research study will correctly detect an effect when it truly exists.

In a recent article, Habibzadeh has proposed a method to compute the most appropriate PST [14]. The method proposed is based on the receiver operating characteristic curve analysis. Using the analogy between diagnostic tests with continuous results and statistical inference tests of hypothesis, the most appropriate PST was computed in the same way as the most appropriate test cut-off value is calculated [14, 15]. Similar to the most appropriate test cut-off value, defined as the value where the weighted sum of errors in making a correct diagnosis is a

minimum [15], the most appropriate PST was defined as the value where the weighted sum of errors in statistical inference is a minimum [14]. It has been shown that the optimum PST ($PST_{opt}$) is not a fixed value for all studies; it depends on various aspects of the study—its sample size, the minimum acceptable effect size, and the prior odds of the alternative hypothesis ($H_1$) relative to the null hypothesis ($H_0$), among other things [14].

In his article, Habibzadeh provided details for a one-tailed Student's $t$ test, as an example. Herein, it is meant to generalize the method proposed to a two-tailed statistical inference test of hypothesis and apply the method to the following hypothetical RCT—a two-arm parallel design RCT where we want to compare the effectiveness of two treatments in two groups of participants (100 in one arm and 70 in another) looking for a medium effect size (Cohen's $d = 0.5$) [16]; assuming three ratios between variances of two study groups of 0.5, 1.0, and 2.0; and a prior odds of $H_1$ relative to $H_0$ of 1.

Herein, the results of using the PST values of 0.05 (the conventional value) [4], 0.005 (the value proposed by Ioannidis and Benjamin) [10, 11], and the optimum value [14] for the PST ($PST_{opt}$, computed according to the method presented in the current study) to interpret the results of the hypothetical RCT and also 22 500 real RCTs retrieved from the CDSR are presented and analyzed.

## Materials and methods

### The most appropriate p value significance threshold

Assume that a cost function is defined to estimate the total error that could happen in statistical inference test of hypothesis—type I ($\alpha$) and type II ($\beta$) errors. Type I error can only happen when $H_0$ is correct (when there is no true effect); type II, when $H_1$ (when there is a true effect). If $pr$ designates the prior probability of $H_1$, then a candidate for the cost function would be:

$$\varepsilon = pr\,\beta + (1 - pr)\,\alpha \tag{1}$$

This is quite similar to the situation with the calculation of the number needed to misdiagnose for a diagnostic test [17], if the $pr$ represents the probability of the disease of interest; $\beta$, the false-negative rate (1 –*Specificity*); and $\alpha$, the false-positive rate (1 –*Sensitivity*) [14]. However, most of researchers believe that the seriousness of making a type II error is not as high as making a type I error. Many scientists, including most of those working in biomedical sciences, assume a maximum acceptable probability of making type I error ($\alpha$) of 0.05, and a minimum acceptable study power (1–$\beta$) of 0.8, hence, a maximum acceptable probability of making a type II error ($\beta$) of 0.2. This implies that our tolerance for making type II error is four times of that making a type I error; in other words, the seriousness of type II error relative to type I error is generally assumed to be 0.25 [16, 18, 19]. It seems reasonable to also weigh different types of errors in computing the cost function with a factor reflecting the seriousness of type II relative to type I error, $C$. The cost function introduced in Eq 1 can thus be written as follows [14]:

$$\varepsilon(t) = C\,pr\,\beta(t) + (1 - pr)\,\alpha(t) \tag{2}$$

where $\alpha$ and $\beta$ are the probability of making type I and type II errors, respectively, given the statistic cut-off value of $t$; and $pr$ designates the prior probability of $H_1$ (Fig 1) [14]. This is very similar to what is used for the computation of weighted number needed to misdiagnose for a diagnostic test [15, 17]. For instance, for a two-tailed Student's $t$ test (the statistical test to be used in the current study), we can write [1, 20]:

$$\alpha(t) = 2\,F(-|t|,\ \nu) \tag{3}$$

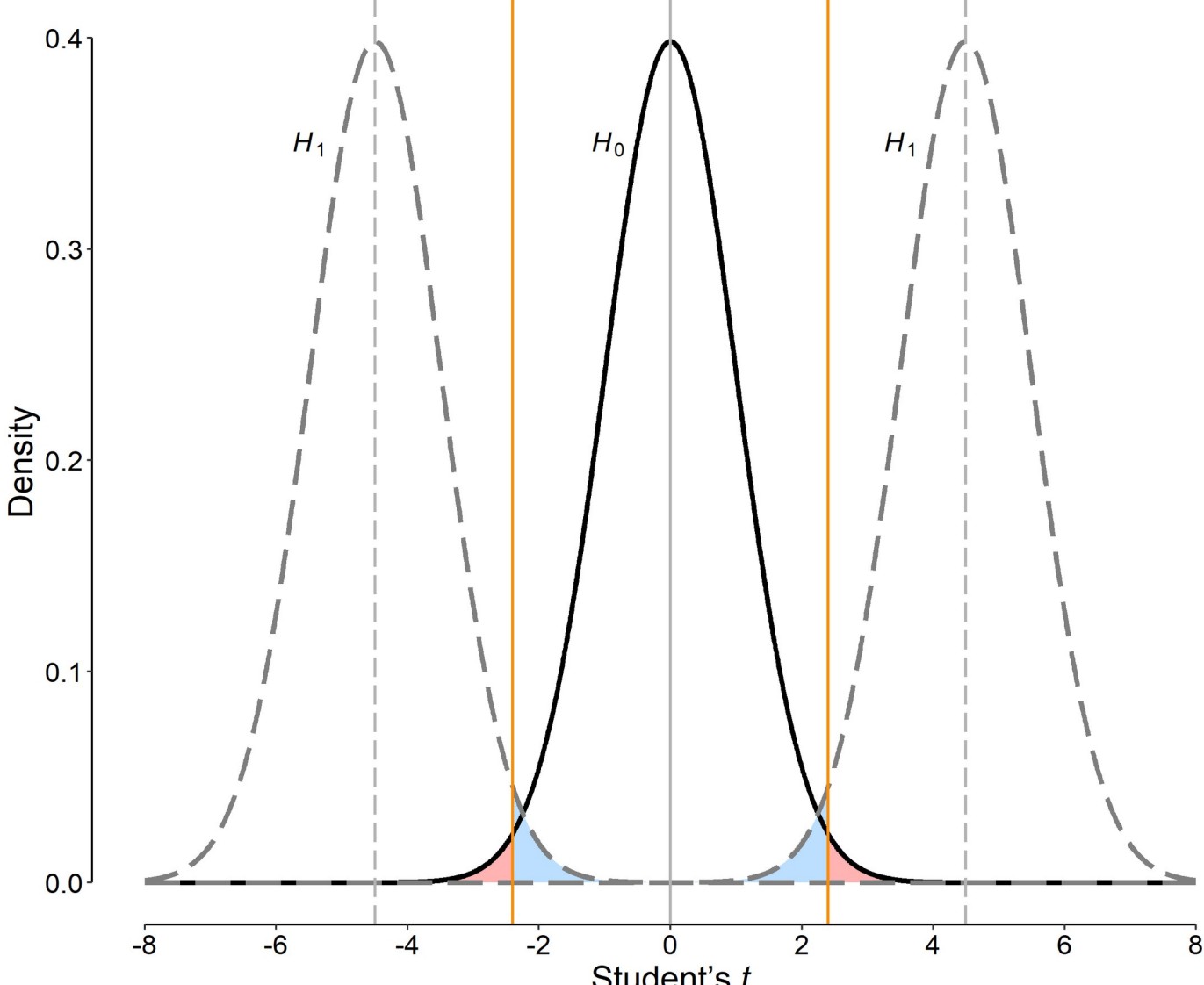

**Fig 1. Distribution of Student's *t* density functions (*i.e.*, the area under each curve is equal to 1) in a two-tailed statistical inference test of hypothesis.** Let assume that under the null hypothesis ($H_0$), the mean value of the statistic is zero (solid curve) and that under the alternative hypothesis ($H_1$), the mean value is non-zero (dashed gray curves, the effect size of interest on *t* scale, $\delta$ in Eq 5). Setting a significance threshold for the statistic of interest (the vertical orange lines) results in two types of errors in the inference—type I error (designated by $\alpha$), to reject $H_0$ while there is no true effect (the probability of which is equivalent to the light red-shaded areas); and type II error (designated by $\beta$), to retain the $H_0$ while there is a true effect (the probability of which is equivalent to the light blue-shaded areas).

and

$$\beta(t) = F(|t| - \delta, \ v) - F(-|t| - \delta, \ v) \qquad (4)$$

where *F* is the cumulative distribution function of Student's *t* distribution and *v*, the degree of freedom [1, 20]. In the current study, the function *pt* from base R was used to compute the *F* (see S1 File). In Eq 4, $\delta$ is the *t* value corresponding to the minimum acceptable effect size, *d*

[14]:

$$\delta = \frac{d\, s_1}{se_\Delta}$$ (5)

where $s_1$ is the standard deviation of data in the first group and $se_\Delta$ is the standard error of the difference of the group means, which is:

$$se_\Delta = \sqrt{s^2\left(\frac{1}{n_1} + \frac{1}{n_2}\right)}$$ (6)

where $n_1$ and $n_2$ are sample sizes of the study groups [14, 21, 22], and $s^2$ is the pooled estimate of the variance and calculated as follows [21]:

$$s^2 = \frac{(n_1 - 1)s_1^2 + (n_2 - 1)s_2^2}{n_1 + n_2 - 2}$$ (7)

where $s_2$ represents the standard deviation of data in the second group. Eq 5 is just rescaling the measure of the minimum acceptable effect size in the first study group with the $s_1$ as the unit of measurement ($d$), into the effect size in the $t$ distribution scale with $se_\Delta$ as the unit of measurement ($\delta$).

In the data set of 22 500 RCTs used in this study, there was no data on $s_1$ and $s_2$. But, there was data on $n_1$, $n_2$, and $se_\Delta$ based on which the pooled estimate of the variance (and the standard deviation, $s$) could be computed (Eq 6). However, there was no way to compute $s_1$, which was necessary for the calculation of $\delta$ (Eq 5), unless $s_1$ and $s_2$ were assumed to be equal. Then, $s$ was equal to $s_1$ and $s_2$ (Eq 7) and after combining Eqs 5 and 6, Eq 5 was simplified to:

$$\delta = \frac{d}{\sqrt{\frac{1}{n_1} + \frac{1}{n_2}}}$$ (8)

Therefore, for analysis of 22 500 RCTs, it was assumed that the study groups had equal variances. Combining Eqs 3, 4, and 8, Eq 2 will be a function of some constants, $C$, $pr$, $\delta$, $n_1$, $n_2$, and $v$, and the variable $t$. The function *optim* (using the method L-BFGS-B) from base *R* was used to numerically calculate the optimum value for $t$ that minimizes the cost function (Eq 2, see S1 File).

## Selection of randomized clinical trials

The data set used for this study was a subset of more than 400 000 records of RCTs retrieved from the CDSR. The data set is publicly available from the Open Science Framework [23]. The criteria used by van Zwet, *et al*, in their studies [1, 20], were also used in the current study to filter the data set. Only RCTs on the efficacy of a treatment with a single (either a dichotomous or continuous) outcome were included. For each study, the primary effect designated by $b$ and its standard error, $se$, were retrieved. van Zwet, *et al*, calculated the $z$ statistic as $b/se$ for studies with continuous outcomes (*e.g.*, change in blood pressure) and $log(b)/se$ for studies with dichotomous outcomes (*e.g.*, odds ratio, relative risk, and hazard ratio). They also excluded RCTs with a $z \geq 20$ [1, 20]. In the current study, the number of participants in each treatment arm, $n_1$ and $n_2$, was also taken into account. Because the number of participants varies widely, only RCTs with a sample size $\geq 10$ were included in the study. Given the low sample size in some studies, instead of $z$ distribution, in the current study, Student's $t$ distribution with a degree of freedom of $v$ was used for interpretation of the results (see S1 File) [24].

## Statistical analysis

*R* software version 4.3.2 (*R* Project for Statistical Computing) was used for data analysis. Cohen's $\kappa$ for binary outcomes, computed with the function *kappa2* from the *R* package *irr* [25], was used for assessing the level of agreement between two raters (Methods). The functions *pt* and *optim*, both from base *R*, were used to compute the value of the cumulative distribution function of Student's *t* distribution, and minimize the error function (Eq 2), respectively. In the current study, a prior odds of $H_1$ relative to $H_0$ of 1 (*i.e.*, a prior probability of 50%, which means that before conducting the RCT, the researchers believed that there was 50% chance that the treatment is truly effective) was assumed for all studies RCTs. This value was based on what has been proposed by Ioannidis and Benjamin [5, 10]. The seriousness of type II error relative to type I error, *C*, was assumed to be 0.25. It was assumed that RCTs with a total sample size $\geq$ 100 were looking for an effect size $\geq$ 0.5; otherwise, because conducting underpowered RCTs is unethical (*e.g.*, conducting an RCT with a total sample size < 100 to detect a small or medium effect), the minimum acceptable effect size was assumed to be 0.8 [16]. Given the sample size in each arm, *v* could then be calculated ($n_1$ + $n_2$−2). Assuming equal variances in study groups, $\delta$ (Eq 8) was then calculated; the optimum *t* value (corresponding to the PST$_{\text{opt}}$) was computed numerically (see S1 File). The probability of type I error ($\alpha$) and type II error ($\beta$) was then calculated from Eqs 3 and 4, respectively. Having $\beta$, the study power was calculated (1–$\beta$). Most scientists in biomedical sciences agree upon a maximum acceptable probability of type I error ($\alpha$) of 0.05; type II error ($\beta$), 0.2 [16, 18, 19]. Imposing these constraints on the computed PST$_{\text{opt}}$ gave the constrained-PST$_{\text{opt}}$ (C-PST$_{\text{opt}}$).

## Ethics

Not applicable: this study was a theoretical study and did not involve any animal or human beings.

## Results

### Hypothetical RCT

Assuming equal variance of the two study arms and using Eq 8 for the calculation of $\delta$, the most appropriate *t* value for a two-sided Student's *t* test for independent groups being used in the hypothetical RCT is 2.26 (Fig 2). This value corresponds to a PST$_{\text{opt}}$ of 0.025 and a study power of 83% (Eq 4; the study power = 1–$\beta$). The weighted sum of errors in statistical inference is a minimum for the PST$_{\text{opt}}$ of 0.025; the error is larger for other PST values, including 0.05 and 0.005 (Fig 2). The error for the PST of 0.005 is even higher than that for the conventional value of 0.05.

If $s_2$, the standard deviation in the second group, is different from $s_1$, for example, $s_2 / s_1 =$ 1.5, then Eq 5 should be used to compute $\delta$. Calculations will then end in completely different values; the most appropriate *t* value for the hypothetical RCT is 2.11; PST$_{\text{opt}}$, 0.037; and study power, 69%. For $s_2 / s_1 = 0.5$, the values are respectively, 2.48, 0.014, and 92%.

The C-PST$_{\text{opt}}$ values for three effect sizes, three prior odds of $H_1$ relative to $H_0$, and different sample sizes are presented in Fig 3. Note that in the calculation of the C-PST$_{\text{opt}}$, it was assumed that the two study arms have the same sample size and variance. For different combinations of sample sizes in each group (yet assuming equal variances), the C-PST$_{\text{opt}}$ are far different from the design with equal sample sizes in each arm (Table 1).

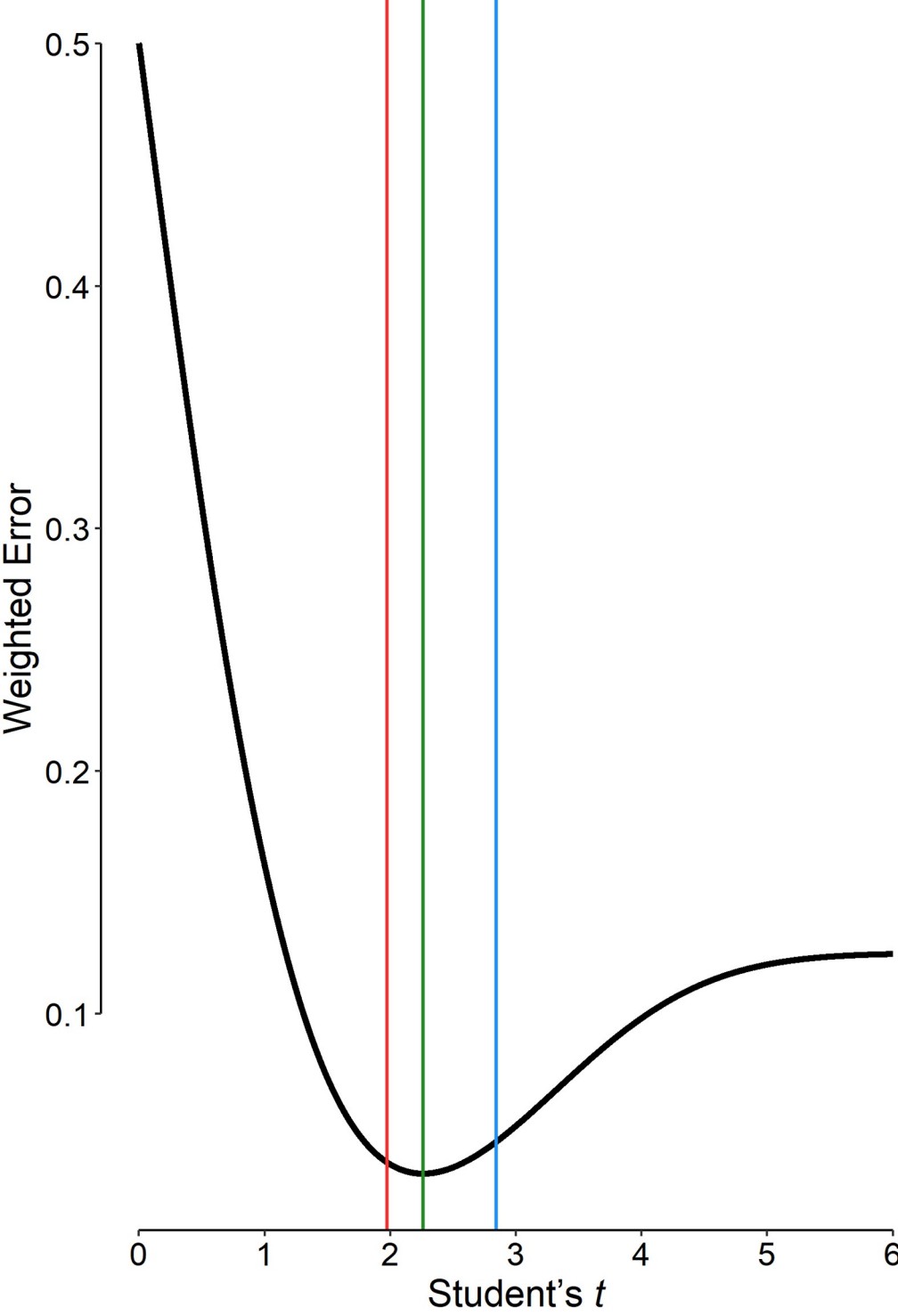

**Fig 2. The amount of weighted error (Eq 2) associated with different values for the Student's *t* cut-off value in our hypothetical randomized clinical trial.** The minimum weighted error corresponds to a *t* value of 2.26 (green line) corresponding to a p value significance threshold of 0.025. The weighted errors associated with other values including the conventional p value significance threshold of 0.05 (red line) and 0.005 (blue line) are larger than the minimum value. In our example, the error associated with the cut-off of 0.005 (blue line) is even larger than that for the cut-off of 0.05 (red line).

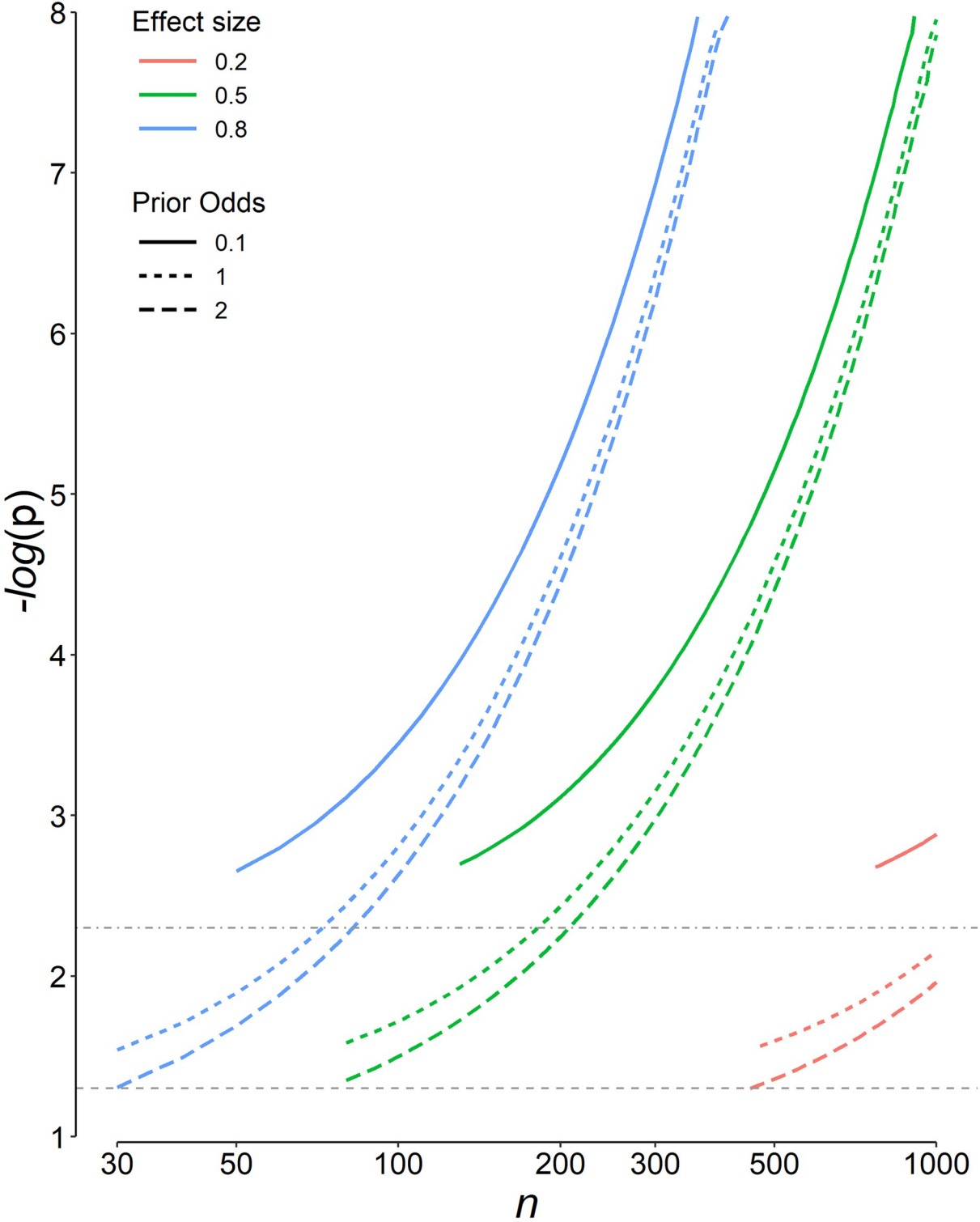

**Fig 3. Variation of the most appropriate p significance threshold after imposing the constraints ($\alpha < 0.05$ and study power $\geq 0.8$), C-PST$_{opt}$, for 3 effect sizes ($d$), 3 prior odds of $H_1$ relative to $H_0$, and different sample sizes in each group ($n$).** Both study arms were assumed to have equal sample size and variances. Only results that with an $\alpha \leq 0.05$ and a study power $\geq 0.8$ are presented (C-PST$_{opt}$). Note that the abscissa has a logarithmic scale. The ordinate is $-log$(C-PST$_{opt}$), *i.e.*, higher values correspond to lower C-PST$_{opt}$. The horizontal dashed gray line corresponds to the conventional p significance threshold of 0.05; dot-dashed, 0.005. For some designs, there is no sample size that works under the constraints imposed on the $\alpha$ and study power. For instance, it is not possible to discover a difference with a minimum

acceptable effect size ($d$) of 0.5 with a sample size of 100 per study group, if a prior odds of $H_1$ relative to $H_0$ of 0.1 is assumed (the solid green line).

### Real RCTs

After applying the inclusion/exclusion criteria, 22 500 RCTs were remained for analysis. The median sample size of the studied RCTs was 82 (range, 10 to 317 400). Taken the conventional PST of 0.05 into account, 28.5% ($n$ = 6407) of RCTs were significant (Fig 4). After decreasing the PST to 0.005, 14.2% ($n$ = 3198) still remained significant. The data set had no information about the $s_1$ and $s_2$. Therefore, it was assumed that variances in two studied groups were equal; Eq 8 was used to compute $\delta$ and the $PST_{opt}$, which revealed that 21.8% ($n$ = 4897) of RCTs were significant (Fig 4). The level of agreement between the latter method and the "p < 0.05" criterion in classifying an RCT to either "significant" or "not significant" was near perfect (Cohen's $\kappa$ = 0.82). None of the RCTs that had a p $\geq$ 0.05 (conventionally considered "not significant"), was found significant using the $PST_{opt}$ criterion; all of the 4897 RCTs that were found significant using the $PST_{opt}$, had a p < 0.05. The $PST_{opt}$ values differed from study to study; the median $PST_{opt}$ for significant RCTs was 3.07 (IQR, 1.93 to 3.90) $\times 10^{-2}$. The study power of significant studies varied from a minimum of 9% to near 100% (median 78%; IQR, 66% to 88%).

Using the $C\text{-}PST_{opt}$ resulted in a situation where only 9.9% ($n$ = 2237) of studied RCTs became significant (Fig 4). Half of the RCTs that were found significant using the conventional PST of 0.05, still remained significant if the cut-off decreased to 0.005; 34.9%, if the $C\text{-}PST_{opt}$ was used. The median $C\text{-}PST_{opt}$ for significant RCTs was 1.91 (IQR, 1.27 to 2.44) $\times 10^{-2}$. The distribution of the study power of significant studies is presented in Fig 5 (median, 88%; IQR, 84% to 92%). If the maximum acceptable type I error was assumed to be 0.005, only 1.4% of the studied RCTs were significant.

## Discussion

The main concern of a scientist is to discover truth about the world. Scientists generate hypothesis, conduct research studies, and examine whether the collected data are consistent with their hypothesis or not. One of the prevailing approaches is using statistical inference

**Table 1. The optimum p value significance threshold after imposing the constraints ($\alpha$ < 0.05 and study power $\geq$ 0.8), $C\text{-}PST_{opt}$, for different combinations of sample size in each arm ($n_1$ and $n_2$) given that the acceptable effect size $\geq$ 0.5, probability of 0.5 ($pr$) that the alternative hypothesis ($H_1$) is correct (odds = 1), and that the seriousness of type II error is one-fourth that of type I error ($C$ = 0.25).**

| $n_1$ | $n_2$ | | | | | |
|---|---|---|---|---|---|---|
| | 50 | 100 | 200 | 300 | 500 | 1000 |
| 50 | — | — | 1.590 | 1.629 | 1.667 | 1.698 |
| 100 | — | 1.719 | 1.956 | 2.077 | 2.200 | 2.313 |
| 200 | 1.590 | 1.956 | 2.434 | 2.723 | 3.054 | 3.401 |
| 300 | 1.629 | 2.077 | 2.723 | 3.153 | 3.690 | 4.310 |
| 500 | 1.667 | 2.200 | 3.054 | 3.690 | 4.574 | 5.748 |
| 1000 | 1.698 | 2.313 | 3.401 | 4.310 | 5.748 | 7.956 |

It is also assumed that the variances in both groups are equal. Note that the values are $-log_{10}(C\text{-}PST_{opt})$. For example, for a study with 100 participants in one arm and 500 in another arm (a total of 600 participants), the $C\text{-}PST_{opt}$ is $10^{-2.2}$, which is 0.0063 meaning that a p value < 0.0063 should be considered "significant." The value is far different from the $C\text{-}PST_{opt}$ for a study with 300 participants in each arm—0.0007 = $10^{-3.153}$. Note that for certain conditions, say 50 participants in one arm and 100 in another, it is not possible to design a study to detect a minimum acceptable effect size of 0.5 with the given constraints.

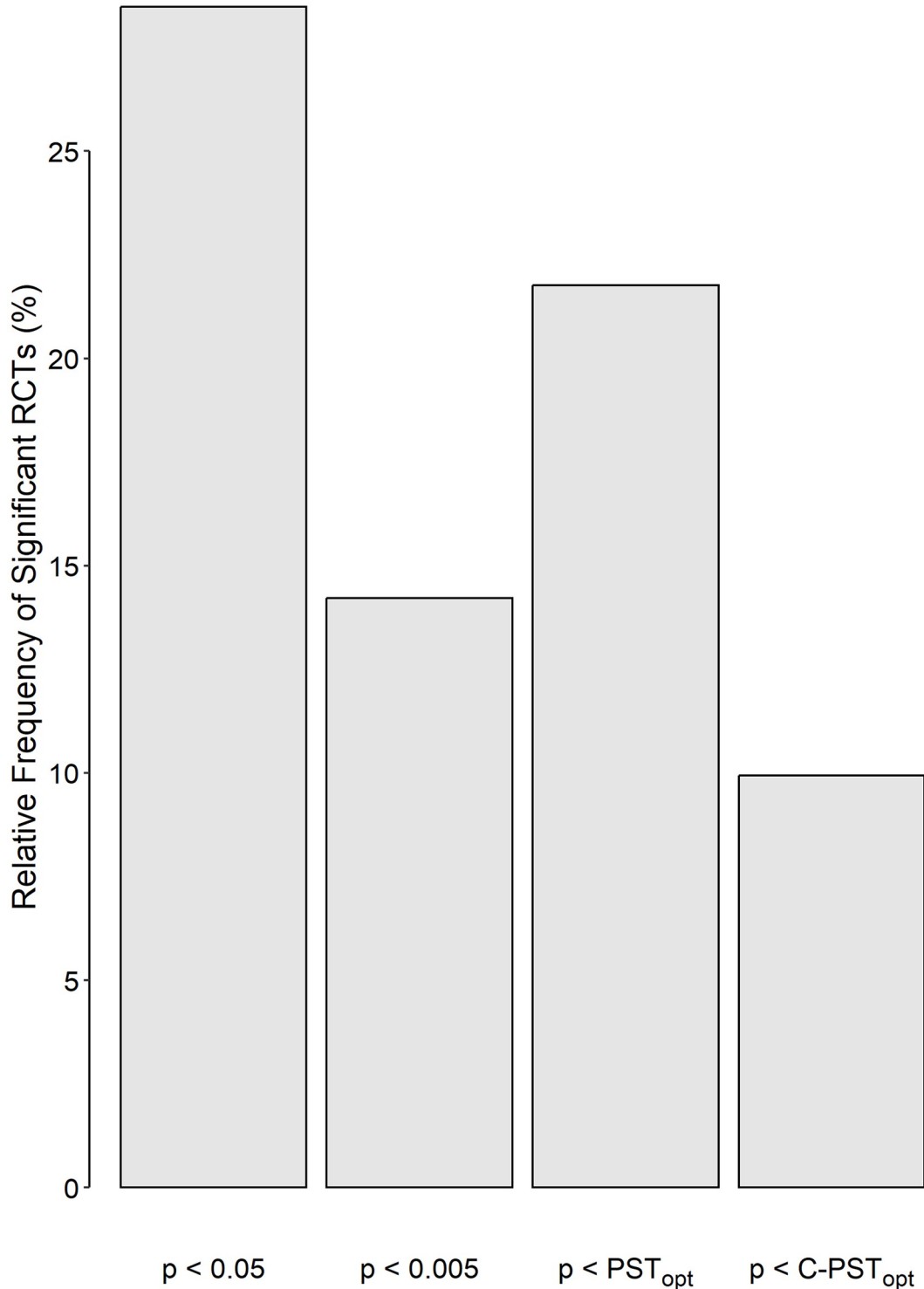

**Fig 4. Frequency distribution of the significant randomized clinical trials (RCTs) using different p value significance thresholds (PSTs) criteria.** The optimum p value significance threshold ($PST_{opt}$) was computed by minimizing Eq 2. $C-PST_{opt}$ is the $PST_{opt}$ after imposing the constraints that a type I error probability of at most 0.05 and a study power of at least 0.8 are mandatory.

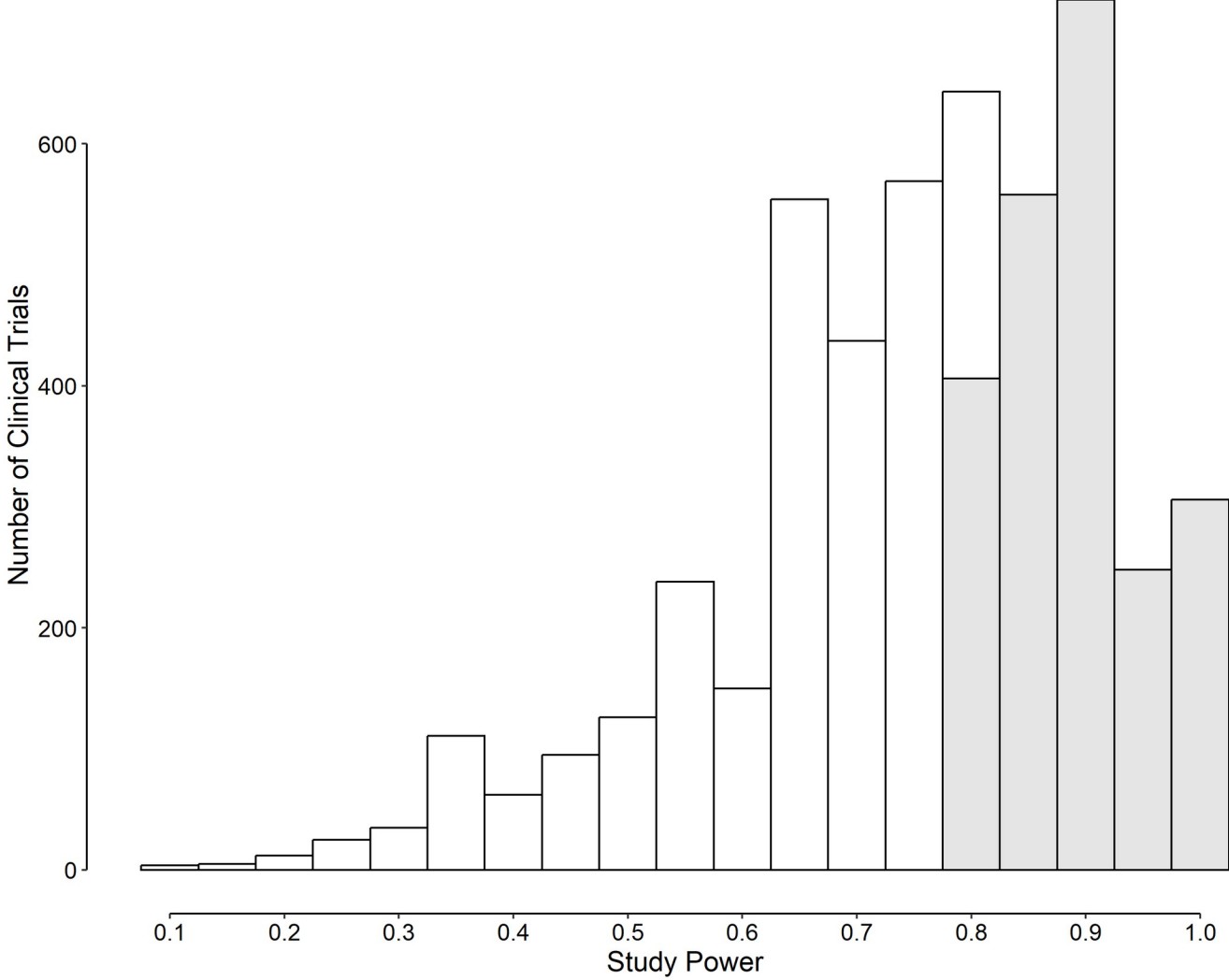

**Fig 5. Frequency distribution of the study power of randomized clinical trials that found significant based on the PST_opt method proposed [14].** The distributions are those before (white columns) and after selecting studies with a type I error probability of at most 0.05 and a study power of at least 0.8 (C-PST_opt, gray columns).

tests of hypothesis [22]. The p value, is the most common single statistic used to determine whether $H_0$ should be rejected or retained. If the calculated p value is less than a set cut-off value (PST), the observed results are considered surprising enough under $H_0$, the results are considered "significant" and the $H_0$ is rejected [14].

No reasonable formulation has so far been provided to determine which value of PST is the best. For instance, Ronald A. Fisher arbitrarily chose a PST of 0.05 in 1925 [4]. This value, although unfounded, has soon become popular, thereafter. A study conducted on more than 350 000 articles published between 1990 and 2015, retrieved from PubMed Central, reveals that there are on average nine p values in each studied article [26].

Simulation studies have shown that using the conventional PST of 0.05 results in many false-positive research results [5, 6]. There is a trade-off between $\alpha$ and $\beta$; a decrease in the PST ($\alpha$), which corresponds to an increase in the Student's $t$ significance threshold, increases $\beta$, and *vice versa* (Fig 1) [27]. Using a lower PST, for instance, by decreasing the PST from 0.05 to

0.005 (as proposed by Ioannidis and Benjamin) [10, 11], although is associated with a lower false-positive rate, is not necessarily associated with a lower weighted sum of type I and type II errors (Eq 2). As an example, for the hypothetical RCT, the weighted sum of errors for the PST of 0.005 is even higher than that for the conventional PST of 0.05 (Fig 2). Therefore, choosing a lower PST does not necessarily improve the situation; false-negative results may occur—*i.e.*, $H_0$ is retained while there is a true effect. In this way, we may overlook important therapeutic effects merely for fear of ending in false-positive results. A fixed PST does not work in all situations; a flexible PST may do [14].

Assuming the conventional PST of 0.05, about 30% of studied RCTs were found significant. For studied RCTs, decreasing the cut-off from 0.05 to 0.005, decreased the significance rate to near 15%. Here again, although the false-positive rate theoretically decreased, there was no guaranty that this was necessarily better; false-negative results could ensue and potentially important therapeutic effects might be left undiscovered. Perhaps, only choosing a flexible PST can optimize the weighted error (Eq 2).

Using $PST_{opt}$, about 20% of studied RCTs were found significant, about two-thirds of those found significant with a conventional PST of 0.05. In fact, the two methods had a near perfect level of agreement in classifying the studied RCTs to either "significant" or "not significant." Although the $PST_{opt}$ values computed in this way, were $< 0.05$ for all significant RCTs, the associated values for the study power varied from a minimum of 9% to near 100% (median 78%; IQR, 66% to 88%). While an $\alpha < 0.05$ is acceptable by many researchers, a study power $< 80$% is not. Many of the RCTs that were found significant using the $PST_{opt}$ criterion had unacceptably low power. Under certain circumstances, the methodology may even end with a $PST_{opt} \geq 0.05$, because there is no constraint on the $\alpha$ and $\beta$ in the calculation of $PST_{opt}$ (it did not happen in the current study).

From a pragmatic point of view, imposing constraints on the $\alpha$ and $\beta$ seems to be mandatory. Traditionally, most researchers in biomedical sciences believe that the maximum acceptable probability of type I error ($\alpha$) is 0.05; type II error ($\beta$), 0.2, hence a minimum acceptable study power ($1–\beta$) of 0.8 [16, 18, 19]. Imposing these constrains to $PST_{opt}$ gives the C-$PST_{opt}$. Using the C-$PST_{opt}$, almost 10% of the studied RCTs became significant. Almost half of the RCTs found significant using the $PST_{opt}$, were discarded with C-$PST_{opt}$ criterion for being under-powered. Employing C-$PST_{opt}$, around two-thirds of RCTs that were found conventionally significant ($p < 0.05$) were considered "not significant." Other constraints may be imposed; for example, if the maximum acceptable $\alpha$ was assumed to be 0.005 (instead of 0.05), only less than 2% of the studied RCTs were found yet significant—seemingly, a too stringent constraint. Therefore, the choice of constraints applied is of paramount importance.

### Optimum p value significance threshold

An inevitable byproduct of using the frequentist statistical inference test of hypothesis is making type I and type II errors. Using a flexible PST to minimize the weighted sum of these errors sounds like a reasonable way to optimally use the method. Use of $PST_{opt}$, however, would end in finding low-powered RCTs "significant." Imposing constraints on the acceptable levels of type I and type II errors and using C-$PST_{opt}$, would solve this problem. Nonetheless, new problems surface.

As the calculation of the cost function (Eq 2) is based on Eqs 3 and 4, which in turn depend on the *F*, the cumulative distribution function of the statistic of interest (in our example, Student's *t* distribution), $PST_{opt}$ should in general be different from statistical test to test, even in a single study. This makes the situation difficult, even unacceptable by many researchers.

To correctly compute the $PST_{opt}$, it is necessary to use Eq 5 to calculate $\delta$. Assuming that the variances of the two study groups are equal, reduces the equation to Eq 8. But, it is not always the case; variances are often different in the two study groups and Eq 8 cannot give a correct estimation of $\delta$. Thus, the final $PST_{opt}$ and C-$PST_{opt}$ would not be accurate, if equal variances assumption is violated. As an example, the $PST_{opt}$ values calculated for variance ratios of 1.5, 1.0, and 0.5 for our hypothetical RCT, were totally different (0.037, 0.025, and 0.014, respectively). As the data set of RCTs did not contain any information on the variance of the studied groups, in the current study, equal variance assumption was made for analyzing the RCTs.

To accurately compute the $PST_{opt}$, Eq 5 should be used. But, Eq 5 cannot be calculated *a priori* because the standard deviations of the two groups will only be available *a posteriori*, after the study is completed [14], which brings us into a fundamental problem.

The variances of data measured in two arms are very likely to be different, even for replicates of a single RCT, for the sampling variation, if nothing else [28]. Therefore, the computed $PST_{opt}$ (based on Eq 5) for replicates of even a single RCT would be very likely to be different [14]. Having different $PST_{opt}$ values, even for a single statistical test, for replicates is not acceptable at all. This dilemma points to an inherent conflict within the frequentist statistical inference framework and makes using the idea of using a PST and the p value a total failure.

Using frequentist methods, the confidence intervals, closely associated with the p value, are also not reliable. Analyzing almost the same RCTs examined in the current study, van Zwet, *et al*, show that the 95% confidence intervals reported therein contained the real value of the statistic of interest in nearly 90% (not the expected 95%) of the time [1].

## Switching the gear

Given the current situation, it seems that it is probably the time to ultimately call for abandoning the p value and instead of dichotomizing RCTs' results into "significant" and "not significant" and reducing the entire data set to a one-dimensional summary measure (*e.g.*, p value), as it is done in the frequentist statistical framework, use another approach (*e.g.*, the Bayesian methods) and try to revise the likelihood of a hypothesis (*e.g.*, if a treatment is more effective than another) in light of the new results obtained from RCTs [29, 30].

Bayesian approach is definitely not perfect. It is commonly more computationally intensive, particularly if the model has numerous variables [14, 29]. However, with the emergence of better computers and application of the artificial intelligence units in various aspects of research, it seems that the time is ripe to switch to Bayesian or other similar methods [31].

## Limitations

One of the limitations of this study was to focus on Student's *t* test, not other statistical tests. Student's *t* test was examined because it was the only test necessary for data analysis in the current research. Another limitation of this study was assuming equal variances in study groups of the 22 500 RCTs studied. More accurate calculations could not possible for lack of information on the variance of each group in the data set used. The assumption of equal variance in RCTs studied was another limitation. Using a cost function other than that used in the current study (Eq 2) would end in different results; other researchers may prefer to use another convex function with other weights. The choice of Eq 2 could thus, be considered another limitation of this study, but the choice was merely based on the analogy between the diagnostic tests and statistical inference test of hypothesis [14]. Future studies should focus on the general form of the problem. Simulation studies should be conducted to examine the changes in false-positive and false-negative rates with different PST criteria.

## Conclusions

Our medical practice is heavily dependent upon the results of RCTs. A significant finding would change the treatment of a disease. A false-positive result would deleteriously affect the health of many people. A false-negative result, on the other hand, would result in deprivation of many suffering people from an effective new treatment. The best way to correctly interpret the results of an RCT, using a frequentist approach, is probably using a way to minimize the weighted sum of errors that may incur in a statistical inference test of hypothesis. The method proposed in the current study to calculate the most appropriate PST, however, ends in an inherent conflict in the frequentist statistical framework. To better interpret the results of an RCT, it seems that another approach, say the Bayesian statistics, needs to be employed. The methods may be computationally intensive, but given the increasing computational capacity of the globe and the introduction of artificial intelligence in most research disciplines, it seems that this is not an important issue to be considered.

## Supporting information

**S1 File. Supplementary materials.** Data dictionary and important pieces of *R* codes. (PDF)

## Acknowledgments

I would like to thank Erik van Zwet, from Leiden University Medical Center, and Simon Schwab, from University of Zürich, for providing parts of the *R* codes for filtering the data (see S1 File) and describing the original data set retrieved from OSF.

## Author Contributions

**Conceptualization:** Farrokh Habibzadeh.

**Data curation:** Farrokh Habibzadeh.

**Formal analysis:** Farrokh Habibzadeh.

**Investigation:** Farrokh Habibzadeh.

**Methodology:** Farrokh Habibzadeh.

**Project administration:** Farrokh Habibzadeh.

**Resources:** Farrokh Habibzadeh.

**Software:** Farrokh Habibzadeh.

**Supervision:** Farrokh Habibzadeh.

**Validation:** Farrokh Habibzadeh.

**Visualization:** Farrokh Habibzadeh.

**Writing – original draft:** Farrokh Habibzadeh.

**Writing – review & editing:** Farrokh Habibzadeh.

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
