## [Decision Letter · Decision Letter 0]

2 May 2024

PONE-D-24-12367Reinterpretation of the Results of Randomized Clinical TrialsPLOS ONE

Dear Dr. Habibzadeh,

Thank you for submitting your manuscript to PLOS ONE. After careful consideration, we feel that it has merit but does not fully meet PLOS ONE’s publication criteria as it currently stands. Therefore, we invite you to submit a revised version of the manuscript that addresses the points raised during the review process.

After careful consideration, we have decided that a major revision of your manuscript is necessary. The reviewers have requested a more comprehensive literature review, clearer justifications and explanations with detailed statistical assumptions, better organization within the statistical methods section, and the inclusion of a discussion on limitations and empirical validation such as a systematic simulation study. Please address these concerns thoroughly in your revision, and clearly detail your changes or provide rationales for any unaddressed comments in a response letter accompanying your revised manuscript.

We look forward to receiving your revised manuscript.

Kind regards,

Teerapon Dhippayom

Academic Editor

PLOS ONE

Journal requirements

Reviewers' comments:

Reviewer's Responses to Questions

**Comments to the Author**

1. Is the manuscript technically sound, and do the data support the conclusions?

Reviewer #1: Yes

Reviewer #2: No

2. Has the statistical analysis been performed appropriately and rigorously? 

Reviewer #1: Yes

Reviewer #2: No

3. Have the authors made all data underlying the findings in their manuscript fully available?

Reviewer #1: Yes

Reviewer #2: Yes

4. Is the manuscript presented in an intelligible fashion and written in standard English?

Reviewer #1: Yes

Reviewer #2: Yes

5. Review Comments to the Author

Reviewer #1: The work is solid. However, please be advised to mention the weaknesses of this study in DISCUSSION. In particular, the crux of the author's argument is based on Eq. (1) whose assumptions might be violated.

Reviewer #2: Review Comments

Summary: This paper proposes an approach selecting appropriate cut-off values for a statistical test and applies it to reanalyze a collection of clinical trials. While it provides some kind of guidance on selecting significance cut-off thresholds, the paper is not well-written, and I do not recognize the necessity of publishing it.

Major comments:

This paper lacks a literature review on the past developments regarding the selection of significance cut-off thresholds for a statistical test. There is not a clear justification provided for using the proposed error function as an objective. Additionally, the components of the objective function are not introduced clearly. For example, while the objective function is a weighted sum of type-I error and type-II error, the weights are not well-defined or interpreted. The objective uses the information on type-II error (or equivalently, power), yet the power is seldom mentioned in the abstract and introduction. It is introduced alongside the objective error function without proper explanations.

The statistical methods section lacks organization. For instance, it would be better to omit the expressions for the distribution function of t-distribution and instead provide explanations for δ before giving the distribution function. In addition, the author fails to introduce statistical assumptions necessary for performing the calculations. For example, it is implicitly assumed that two groups are independent and normally distributed with the same variance. I do not think the current level of rigor is sufficient for introducing a completely new approach without acknowledging previous developments. In addition, using t-test only may not be always appropriate.

The author employs overly definitive language, which appears very weak to me. For example, the author states, ‘Frequentist statistics has internal conflict. …’ (line 31-33), yet little evidence is given to support why the Bayesian statistics is better than the frequentist statistics, even in this specific question. I strongly disagree with making such assertions without giving sound evidence because the methods proposed in this paper do not use any knowledge or setup from Bayesian’s framework.

This paper only proposes a new method without giving any discussions on the limitations or drawbacks. For example, using the proposed objective function depends on tuning multiple parameters and minimizing an objective involving the power may not be the best way of controlling false-positive.

There may be many other interesting aspects relevant to the research question in this paper. For example, the impact of choosing weights on the resulting optimal cut-off values, and the relationship between parameters and final optimizer (e.g., monotone or quadratic). Why the objective is a weighted sum instead of other convex shapes? The current statistics is t-test and what is the consequence of using other test statistics?

Minor comments:

In the background of the abstract, the author states, ‘Many researchers show that results of RCTs are mostly false-positive’ (line 15-16). In my opinion, a false-positive is defined as the event incorrectly rejecting the null hypothesis. One cannot identify a false-positive without knowing beforehand that the null hypothesis is true. Statistically speaking, a false-positive is a binary random variable with a prespecified probability α, known as type-I error rate, which is a crucial quantity that a statistician would like to control.

In the discussion section (line 231-244), the author argues for viewing the parameters as a posteriori instead of priori. I think there is gap between current methods and methods capable of handling parameters as a posterior. I suggest filling in this blank in the methods section.

In the discussion section (line 255-258), the author mentions that the computational intensity may be a drawback when using the Bayesian’s framework. However, in my opinion, many modern computers can already handle the computational task if the model is not overly complex, e.g., the levels of the data are not too large.

This paper lacks a systematic simulation study to demonstrate the control of false-positives and false-negatives.

6. PLOS authors have the option to publish the peer review history of their article (what does this mean?). If published, this will include your full peer review and any attached files.

Reviewer #1: **Yes: **Junren Chen

Reviewer #2: No

---

## [Author Response · Author response to Decision Letter 0]

12 May 2024

Dear Editor,

Thank you very much for informing me of the fruitful comments and suggestions made by respected editorial team and the respected referees. I revised the manuscript accordingly. I believe the revised version is much better and clearer compared with the original one. I hope you will find the revised version acceptable for publication in your esteemed Journal, PLoS ONE.

Best Regards,

F. Habibzadeh, MD

Editor’s Comments

After careful consideration, we have decided that a major revision of your manuscript is necessary. The reviewers have requested a more comprehensive literature review, clearer justifications and explanations with detailed statistical assumptions, better organization within the statistical methods section, and the inclusion of a discussion on limitations and empirical validation such as a systematic simulation study. Please address these concerns thoroughly in your revision, and clearly detail your changes or provide rationales for any unaddressed comments in a response letter accompanying your revised manuscript.

Response: Thank you very much for your valuable comments. Literature review has been done. The Introduction was updated. I also tried to revise the manuscript based on referees’ comments. The simulation was not conducted as its results would form a separate original article, not a part of the current manuscript.

Reviewer #1:

The work is solid. However, please be advised to mention the weaknesses of this study in DISCUSSION. In particular, the crux of the author's argument is based on Eq. (1) whose assumptions might be violated.

Response: Thank you very much for raising this important issue. Limitations of the current study (studying only Student’s t test and using the current cost function (Eq. 2 [in the revised version]) and other limitations) were inserted in the Discussion.

Reviewer #2:

Review Comments

Summary: This paper proposes an approach selecting appropriate cut-off values for a statistical test and applies it to reanalyze a collection of clinical trials. While it provides some kind of guidance on selecting significance cut-off thresholds, the paper is not well-written, and I do not recognize the necessity of publishing it. 

Response: Thank you very much for your comments. The structure of the manuscript was improved.

Major comments:

 This paper lacks a literature review on the past developments regarding the selection of significance cut-off thresholds for a statistical test.

Response: Thank you very much for your valuable comments. Literature review has been done. The Introduction was updated.

There is not a clear justification provided for using the proposed error function as an objective. Additionally, the components of the objective function are not introduced clearly. For example, while the objective function is a weighted sum of type-I error and type-II error, the weights are not well-defined or interpreted. The objective uses the information on type-II error (or equivalently, power), yet the power is seldom mentioned in the abstract and introduction. It is introduced alongside the objective error function without proper explanations. 

Response: Thank you very much for your great comments and suggestions. Statements describing the rationale behind using the cost (objective) function were added to the Introduction, Methods and Abstract. The rationale behind choosing the weights were also explained. Most of these were bases on the analogy existing between the diagnostic tests and statistical inference tests of hypothesis (Ref 14).

 The statistical methods section lacks organization. For instance, it would be better to omit the expressions for the distribution function of t-distribution and instead provide explanations for δ before giving the distribution function. 

Response: Thank you very much for your great comments. The expression for the distribution function of t-distribution was omitted. δ is explained in more detail; it is the minimum acceptable effect size, d (measured as s1 [the standard deviation of data in the first group] as the unit), rescaled to the t-distribution scale with the standard error of difference being as the unit of measurement.

In addition, the author fails to introduce statistical assumptions necessary for performing the calculations. For example, it is implicitly assumed that two groups are independent and normally distributed with the same variance. I do not think the current level of rigor is sufficient for introducing a completely new approach without acknowledging previous developments. In addition, using t-test only may not be always appropriate. 

Response: Thank you very much for raising this important issue. In fact, this is not a completely new approach. I have presented the original work in a previously published article (see Reference 14, doi: 10.1186/s12967-023-04827-8). Herein, I just meant to extend the work to two-tailed statistical tests and apply the method to 22 500 RCTs. In this work Student’s t test was used. The assumptions that the “two groups are independent and normally distributed with the same variance” are the basic assumptions of Student’s t test. In this study the focus was made to Student’s t distribution, because examining the RCTs studied was based on the Student’s t test. Finally, it was found that accurate computation of the most appropriate p cut-off value needs knowledge about the study results, a posteriori (Eq. 5), which results in an internal conflict in frequentist statistical approach, as I described in the manuscript. Technically, that’s enough to show that a system has an internal conflict with a certain test, a counterexample! 

 The author employs overly definitive language, which appears very weak to me. For example, the author states, ‘Frequentist statistics has internal conflict. …’ (line 31-33), yet little evidence is given to support why the Bayesian statistics is better than the frequentist statistics, even in this specific question. I strongly disagree with making such assertions without giving sound evidence because the methods proposed in this paper do not use any knowledge or setup from Bayesian’s framework.

Response: Thank you very much for raising this very important issue. The issue of the inherent conflict was described in more detail. I do not have much concern about using Bayesian approach. I just mention it as a well-known substitute for many researchers. If one accepts that, like in case of diagnostic tests with continuous results where we cannot use a constant cut-off for all tests, we cannot also use a fix p cut-off value for all studies, then an inherent conflict will surface if we’re going to use the approach I presented here (using the cost function mentioned in Eq. 2). This necessitates accessing the study results to compute the most appropriate p cut-off value, a posteriori. Given the sampling error present in every research, even replicates, the results of two replicates will be slightly different; different p value cut-offs will be computed, which is technically, not acceptable for replicate studies; the results in one replicate may be “significant;” in another, “non-significant.” I tried to address this issue more clearly in the Discussion.

 This paper only proposes a new method without giving any discussions on the limitations or drawbacks. For example, using the proposed objective function depends on tuning multiple parameters and minimizing an objective involving the power may not be the best way of controlling false-positive. 

Response: Thank you very much for raising this important issue. Limitations of the current study (studying only Student’s t test and using the current cost function (Eq. 2 [in the revised version]) and other limitations) were added to the Discussion.

 There may be many other interesting aspects relevant to the research question in this paper. For example, the impact of choosing weights on the resulting optimal cut-off values, and the relationship between parameters and final optimizer (e.g., monotone or quadratic). Why the objective is a weighted sum instead of other convex shapes? The current statistics is t-test and what is the consequence of using other test statistics?

Response: Thank you very much for pointing out this issue. These were addressed in the Discussion as limitations of the current study.

Minor comments:

 In the background of the abstract, the author states, ‘Many researchers show that results of RCTs are mostly false-positive’ (line 15-16). In my opinion, a false-positive is defined as the event incorrectly rejecting the null hypothesis. One cannot identify a false-positive without knowing beforehand that the null hypothesis is true. Statistically speaking, a false-positive is a binary random variable with a prespecified probability α, known as type-I error rate, which is a crucial quantity that a statistician would like to control. 

Response: Thank you very much for raising this important issue. I very much agree with you. The statement was changed. In several simulations (where we are aware if there is a true effect or not) it has been shown that there are many false-positive results with the conventional p cut-off value of 0.05. An example is the study of Ioannidis 2005 PLoS Medicine article (Ref 5).

 In the discussion section (line 231-244), the author argues for viewing the parameters as a posteriori instead of priori. I think there is gap between current methods and methods capable of handling parameters as a posterior. I suggest filling in this blank in the methods section.

Response: Thank you very much for raising this important issue. Sorry for not being clear. I have addressed the point in the Methods and Discussion. To accurate calculation of the optimum p significance threshold, it is necessary to be aware of the variances in the two study groups, which is generally different in the two study groups; the assumption of equality of variance may generally not be correct. This means that accurate computation of the cut-off needs information about the variances, which are only available a posteriori. Otherwise, equality of variances should be assumed which ends to an estimated cut-off value.

 In the discussion section (line 255-258), the author mentions that the computational intensity may be a drawback when using the Bayesian’s framework. However, in my opinion, many modern computers can already handle the computational task if the model is not overly complex, e.g., the levels of the data are not too large. 

Response: Thank you very much for your comment. I do completely agree with you. I mentioned that the Bayesian methods need more calculations, particularly in models with many variables, but, given the current computers, there should be no worry about this issue.

 This paper lacks a systematic simulation study to demonstrate the control of false-positives and false-negatives. 

Response: Thank you very much for your suggestion. I agree that conducting a simulation would make the results clearer, but, putting also the results of a simulation would provide a large amount of information that is more than the acceptable amount of information generally expected in an original article. 

 

PLOS-review-v02.1

The author chose C = .25 (line 133) based on the Handbook of Clinical Psychology. Is this justifiable when analyzing a broad range of RCTs that are mostly not related to psychology?

Response: Thank you for your raising this issue. The value is applicable in many scientific disciplines. References were also cited for biomedical sciences. In fact, the maximum tolerable α and β in clinical trials (and many other study designs in biomedical sciences) are 0.05 and 0.2, respectively.

Line 88: The phrase ‘relative to H_0’ should be deleted.

Response: Thank you for your comment. You’re correct. It was deleted.

Line 97: Instead of ‘the effect size of interest’, using ‘the minimum acceptable effect size’ is more accurate.

Response: Thank you for your suggestion. It was changed.

Line 106 Suggest to re-phrase as ‘R can be used’

Response: Thank you for your suggestion. I believe the statement is clearer as it is. I leave it to editors to change it if they deem appropriate.

Definition of C is inconsistent. Line 133 “C, the seriousness of type I error relative to type II error, was assumed to be 0.25” and Line 87 “C represents the relative seriousness of β compared to α”

Response: Thank you very much for pointing out this mistake. C is the seriousness of type II error relative to type I error. It was corrected in both places.

Line 138: Suggest to re-phrase as ‘were looking for an effect size ≥ .5’

Response: Thank you for your suggestion. It was changed.

Line 140: Suggest to re-phrase as ‘were looking for an effect size ≥ .8’

Response: Thank you for your suggestion. It was changed.

Why does author choose α = .05 and β = .2 when calculating C-PST_opt.

Response: Thank you very much for raising this question. An α of at most 0.05, and a β of at most 0.2 (corresponding to a study power of at least 0.8) are the common values used in biomedical sciences. It seems that in many scientific disciplines, at least in biomedical sciences, type I and type II errors more than these values are not tolerated. That’s the reason these values were selected. Of course, other values may be chosen as constrains.

A few graphs that show how PST_opt varies with C, pr, n1, n2 and the minimum acceptable effect size would be helpful.

Response: Thank you for your suggestion. A Figure was added.

Adding a table that displays values of PST_opt (assuming C = .25, pr = .5 and d = .5 or .8) at different n1 and n2 values would be useful to other researchers who might also want to apply the author’s method.

Response: Thank you for your suggestion. Table 1 was inserted.

Line 204 ‘Fig 1’ should be ‘Fig 2’

Response: Thank you very much for pointing out this mistake. It was corrected.

Line 210: Suggest to re-phrase as ‘decreased by about 15 percentage points’

Response: Thank you for your suggestion. It was changed.

The author should mention in DISCUSSION the weaknesses of Eq. (1), the crux of his/her argument. Eq. (1) assumes all negatives have zero effect and all positives have an effect size that is equal to the minimum acceptable value. The reality is very different and certainly not binary. Another potential problem is that the author implicitly assumes the true size d of a positive effect depends on the cohort size of the study. This is of course incorrect. True effect size is an objective, worldly phenomenon regardless of how a trial is conducted.

Response: Thank you for raising this issue. The limitations of the work were presented in the Discussion. Furthermore, it is not assumed that negative values have “zero effect.” The absolute values used in Equations 3 and 4 are because of the symmetry exists in the Student’s t distribution (Fig 1). The functions for α and β (and also study power) I used in the R codes (Supplementary materials) as well as Eqs 3 and 4, work perfectly for both positive and negative values of t (and of course for t = 0).

---

## [Decision Letter · Decision Letter 1]

24 May 2024

Reinterpretation of the results of randomized clinical trials

PONE-D-24-12367R1

Dear Dr. Habibzadeh,

We’re pleased to inform you that your manuscript has been judged scientifically suitable for publication and will be formally accepted for publication once it meets all outstanding technical requirements.

Kind regards,

Teerapon Dhippayom

Academic Editor

PLOS ONE

Additional Editor Comments (optional):

Both reviewers were satisfied with your responses ant thought the revised manuscript met the standards for publication. I believe this is an interesting topic that will generate further discussion to improve the quality and application of RCTs, and I would like to thank you for your contributions in this field. 

Reviewers' comments:

Reviewer's Responses to Questions

**Comments to the Author**

1. If the authors have adequately addressed your comments raised in a previous round of review and you feel that this manuscript is now acceptable for publication, you may indicate that here to bypass the “Comments to the Author” section, enter your conflict of interest statement in the “Confidential to Editor” section, and submit your "Accept" recommendation.

Reviewer #1: All comments have been addressed

Reviewer #2: All comments have been addressed

2. Is the manuscript technically sound, and do the data support the conclusions?

Reviewer #1: Yes

Reviewer #2: Yes

3. Has the statistical analysis been performed appropriately and rigorously? 

Reviewer #1: Yes

Reviewer #2: Yes

4. Have the authors made all data underlying the findings in their manuscript fully available?

Reviewer #1: Yes

Reviewer #2: Yes

5. Is the manuscript presented in an intelligible fashion and written in standard English?

Reviewer #1: Yes

Reviewer #2: Yes

6. Review Comments to the Author

Reviewer #1: (No Response)

Reviewer #2: The author has carefully addressed my comments. In summary, the introduction and presentation of the methodology have been significantly improved and clarified. The language has been softened and now reads appropriately. Additionally, the limitations have been thoroughly discussed. I believe this version is well-written, with clear interpretations and a sufficient literature review of past developments.

7. PLOS authors have the option to publish the peer review history of their article (what does this mean?). If published, this will include your full peer review and any attached files.

Reviewer #1: No

Reviewer #2: No

---

## [Editor Report · Acceptance letter]

5 Jun 2024

PONE-D-24-12367R1 

PLOS ONE

Dear Dr. Habibzadeh, 

I'm pleased to inform you that your manuscript has been deemed suitable for publication in PLOS ONE. Congratulations! Your manuscript is now being handed over to our production team.

Kind regards, 

on behalf of

Dr. Teerapon Dhippayom 

Academic Editor

PLOS ONE